# Recurrent UTI: Questions and Answers on Clinical Practice

**Tommaso Cai** [1,2,*] , **Massimiliano Lanzafame** [3] **and Carlo Tascini** [4]

1   Department of Urology, Santa Chiara Regional Hospital, 38123 Trento, Italy
2   Institute of Clinical Medicine, University of Oslo, 0010 Oslo, Norway
3   Department of Infectious Diseases, Santa Chiara Regional Hospital, 38123 Trento, Italy
4   Department of Medicine (DAME), Infectious Diseases Clinic, University of Udine, 33100 Udine, Italy
*   Correspondence: ktommy@libero.it; Tel.: +39-0461-903306

**Abstract:** Recurrent urinary tract infection (rUTI) management is still a challenge due to the lack of a standard approach and due to the burden of diseases both on personal and societal aspects. Consultations for rUTIs in everyday clinical practice range from 1% to 6% of all medical visits with high social and personal associated costs, such as prescriptions, hospital expenses, days of sick leave due to the disease, and the treatment of related comorbidities. Recurrent UTIs are, then, associated with anxiety and depression due to treatment failures and symptomatic recurrences. Often urologists are asked to give practical recommendations to patients regarding the everyday management of recurrent UTIs. Here, we aim to give to the physicians managing UTI some helpful suggestions for their everyday clinical practice, on the basis of the recent evidence.

**Keywords:** urinary tract infections; antimicrobial stewardship; antibiotic resistances; quality of life; burden of diseases



## 1. Background and Aims

Uncomplicated urinary tract infections (UTI) are one of the most common infectious diseases in women with a high impact on patients' quality of life [1,2]. In particular, recurrent UTI (rUTI) is related to anxiety and depression due to treatment failures and symptomatic recurrences [3]. Moreover, several reports showed that medical visits for rUTIs in everyday clinical practice range from 1% to 6% of all medical visits with high social and personal associated costs, such as prescriptions, hospital expenses and days of sick leave due to the disease [4,5]. In the last years, many steps forward have been accomplished for understanding the natural history and management of rUTI, even if we are so far from deep comprehension [6]. On the other hand, in the last few decades, the internet has become an easily accessible source of medical information for patients and in particular for rUTI [7,8]. Patients generally tend to use the web to find therapeutic suggestions and new treatments for treating acute episode and preventing recurrent symptomatic episodes. However, several times the patients found uncertain news and, often, dangerous advice on the management of the disease. In this sense, all physicians managing UTIs are required to be informed about the newest evidence in rUTI management. Often urologists are asked to give a practical recommendation to patients regarding the everyday management of recurrent UTI. The aim of this Editor's perspective paper is to give a brief narrative review of the management of rUTIs in order to provide the readers with some suggestions to use in everyday clinical practice, on the basis of the recent evidences, by using an easy and rapid to consult questions and answers form.

## 2. Materials and Methods

*Search of Evidences*

This review aims to update recurrent UTI management focusing on the risk factors identification and new therapeutic strategies to use in everyday clinical practice in light

of antimicrobial stewardship. A search was performed in PubMed, Cochrane CENTRAL and Scopus databases for pertinent papers by using the following terms "recurrent urinary tract infections" AND "antimicrobial stewardship" AND/OR "risk factors" AND/OR "treatment" in line with our previous paper [9]. All references cited in relevant articles were also reviewed and analyzed. The filters used included the English language; and humans. Even if this search was planned as a narrative review, our search was performed in line with the Preferred Reporting Items for Systematic Reviews and Meta-analyses (PRISMA) and the recommendations of the European Association of Urology Guidelines (EAU) office for conducting systematic reviews and meta-analyses [10,11]. We identified 100 potential articles. After the first screening round, 19 articles were considered eligible for inclusion in this narrative review (Table 1).

**Table 1.** The table shows the summary of all included studies in this narrative review. AUA: American Urological Association; CUA: Canadian Urological Association; SUFU: Society of Urodynamics, Female Pelvic Medicine and Urogenital Reconstruction; rUTI: recurrent Urinary Tract Infections; Infectious Diseases Society of America; QoL: Quality of life; LUTIRE nomogram: Lower Urinary Tract Infection Recurrent risk nomogram.

| Author | Year | Type of Study | Aim | Findings Description |
|---|---|---|---|---|
| Anger J. [3] | 2019 | AUA/CUA/SUFU Guideline | Recommendations on rUTI in women | Recommendations on rUTI management. |
| Nicolle LE. [4] | 2005 | IDSA Guideline | Recommendations on asymptomatic bacteriuria | Recommendations on asymptomatic bacteriuria management. |
| Medina M. [5] | 2019 | Narrative review | Evaluate the epidemiology, burden of rUTIs and actual management. | rUTIs are related to high prevalence, and high social and economic impact. |
| Cai T. [6] | 2022 | Prospective study | Evaluate the impact of risk factors evaluation on the natural history of rUTI. | Risk factor identification and counseling may change the natural history of recurrent urinary tract infections, reducing the number of symptomatic episodes, antibiotic usage, and improving patients' quality of life. |
| Cai T. [9] | 2022 | Review | Evaluate UTI management during the COVID-19 pandemic | During the COVID-19 pandemic, all physicians are asked to maintain a high level of adherence to antimicrobial stewardship. |
| Naber KG. [12] | 2022 | Review | Evaluate the psychosocial burden of rUTI | The psychosocial burden of rUTI seems high in everyday clinical practice but little data are available. |
| Cai T. [13] | 2021 | Review | Evaluate the aspects of the patient's quality of life in urology | Quality of life assessment is mandatory in the management of patients affected by rUTI. |
| Wagenlehner F. [14] | 2018 | Prospective study | Web-based survey in 5 countries (Germany, Switzerland, Poland, Russia, and Italy), on women affected by rUTI. | rUTIs have a significant impact on the QoL of women in Europe. |
| Bonkat G [15] | 2022 | EAU Guidelines | Recommendations on rUTI in women | Recommendations on rUTI management. |
| Cai T. [16] | 2012 | Randomized study | Evaluate the impact of asymptomatic bacteriuria treatment on the recurrence rate in rUTI | Asymptomatic bacteriuria should not be treated in rUTI. |

**Table 1.** *Cont.*

| Author | Year | Type of Study | Aim | Findings Description |
|---|---|---|---|---|
| Cai T. [17] | 2015 | Longitudinal cohort study | Evaluate the impact of asymptomatic bacteriuria treatment on antibiotic resistance in rUTIs | Asymptomatic bacteriuria is associated with a higher occurrence of antibiotic-resistant bacteria. |
| Cai T. [18] | 2014 | Prospective study | Development and validation of a nomogram. | LUTIRE nomogram is able to predict the risk of a new symptomatic episode in women with rUTI. |
| Stapleton A. [19] | 1992 | Observational study | To investigate the hypothesis that blood group secretor status is associated with a higher risk of rUTI. | The blood group's non-secretor status increased susceptibility to recurrent UTI. |
| Cai T. [20] | 2021 | Systematic review and Meta-analysis | Evaluate the effectiveness and safety profile of xyloglucan, hibiscus, and propolis in rUTI | Xyloglucan, hibiscus, and propolis are superior to comparator regimens in terms of microbiological and clinical efficacy in rUTI. |
| Camilleri M. [21] | 2012 | Review | The role of the intestinal barrier in rUTI | Intestinal barrier function has a pivotal role in the genesis of rUTI. |
| Guglietta A. [22] | 2017 | Review | Risk factors evaluation in rUTI. | Risk factors evaluation is an important step in the management of rUTI. |
| Esposito E. [23] | 2018 | Animal model | Evaluate the intestinal barrier effect of xyloglucan in rats. | Xyloglucan shows a protective barrier properties in the prevention of UTI in an animal model. |
| Fraile B. [24] | 2017 | In vitro study | Evaluate the role of Xyloglucan in the prevention of urinary infections. | Xyloglucan shows a nonpharmacological barrier property for the management of urinary tract infections. |
| Costache RC. [25] | 2019 | Prospective study | Xyloglucan versus placebo in UTI management. | Xyloglucan + gelose is able to reduce bacteriological and symptomatic parameters in women with rUTI. |

## 3. Results and Evidences

The findings of this systematic review are displayed as clinical questions and answers in order to be utilizable and easy to use in everyday clinical practice.

### 3.1. Which Is the Burden of Uncomplicated Urinary Tract Infections?

UTI are the most common outpatient infections in the United States (US) with an epidemiological spike in young women [3,4]. The peak rate of uncomplicated UTIs occurs usually between the ages of 18 and 39 [5]. After the first episode of a UTI, 27% of women have a symptomatic recurrence within the next 6 months, and 2.7% have a second recurrence within the same period of time. Due to recurrences and symptoms, UTIs are an important cause of quality-of-life impairment [5]. For this reason, quality of life should be an important aspect to evaluate when managing patients affected by UTI. Moreover, it is demonstrated that rUTI is associated with a reduced quality of life in terms of social relationships, self-esteem, and capacity for work [12]. On the basis of this evidence, the burden of uncomplicated UTI, in terms of psychological and social impact, should be considered in everyday clinical practice [13].

### 3.2. What Is the Psychological Burden of rUTI on the Patient's Outcome and on the Adherence to the Treatment?

Recently, an anonymous, self-administered web-based survey has been conducted in five countries on adult women who had experienced recurrent UTI (GESPRIT study),

reporting that rUTI has a significant impact on patients' quality of life in Europe [14]. Moreover, Cai et al. demonstrated that accurate risk factor identification and counseling may change the natural history of rUTI and reduce the number of symptomatic episodes, reduce antibiotic usage, and improve patients' quality of life, reducing patients' stress and anxiety levels [6]. Even if the patients' psychological burden of rUTI is high, up to the moment no studies have been performed on the psychological burden of rUTI on the patient's outcome and on adherence to the treatment. However, in everyday clinical practice, the possible impact of the psychological burden of rUTI on the patient's outcome should be hypothesized.

### 3.3. What Is the Role of Uncomplicated UTI Management in Antimicrobial Stewardship Programs?

Several times, antimicrobials are prescribed inappropriately in outpatients affected by uncomplicated UTI [9,15]. Antimicrobial use, especially when inappropriate, is associated with the selection of antimicrobial-resistant organisms colonizing or infecting the urinary tract [3]. All physicians managing uncomplicated UTI are asked to consider the principles of antimicrobial stewardship. In other words, the antibiotic treatment of UTI should be performed considering the principles of antimicrobial stewardship [15]:

- antibiotic treatment should be used in case of urinary symptoms and absence of vaginal infection
- antibiotic selection, antibiotic dosage, and time schedule should be selected in line with international guidelines recommendations
- treat asymptomatic bacteriuria only in pregnant women and before urological procedures

### 3.4. Is the Presence of Bacteria in the Urine Always a Symptom of Infection and Does It Need to Be Treated?

The only presence of bacteria in the urine is not necessarily a symptom of infections. Indeed, a lot of bacteria are often found in the urine of asymptomatic subjects, being such a condition defined as asymptomatic bacteriuria. Asymptomatic bacteriuria should be only treated in pregnant women and in all those subjects that must undergo endourological interventions [15]. In all other cases, treatment of asymptomatic bacteriuria is absolutely useless and even dangerous for the development of resistance. In patients affected by recurrent UTI, the asymptomatic bacteriuria treatment is, then, not indicated. Cai et al. demonstrated in two clinical trials that asymptomatic bacteriuria treatment is associated with a higher risk of symptomatic recurrence and a higher risk of bacterial resistance [16,17]. In this sense, appropriate counseling about the benefit and harms of the treatment of asymptomatic bacteriuria should be performed in everyday clinical practice.

### 3.5. If, after an Episode of Acute Cystitis, Urine Culture Is Still Positive Must the Antibiotic Treatment Be Repeated?

If, after the antibiotic treatment, the patient has no more symptoms but only bacteria in the urine (asymptomatic bacteriuria) it is absolutely not necessary to repeat the treatment. Indeed, the treatment of asymptomatic bacteriuria in rUTI is harmful and dangerous because it can favor a shorter-term relapse and can stimulate the development of relapses. Please consider the symptoms and not the bacterial growth to start the antimicrobial treatment [15].

### 3.6. Is the Execution of Urine Culture Always Necessary before Prescribing an Antibiotic in One Patient with Symptoms to Be Referred to as Cystitis?

No. Indeed, the management of a patient with symptoms to refer to acute cystitis considers the use of empirical and reasoned antibiotic therapy [15]. This approach involves an accurate assessment of risk factors and knowledge of local resistant bacteria [6,15]. EAU guidelines suggest starting with a 3-g dose of fosfomycin trometamol or, alternatively, nitrofurantoin 100 mg twice daily for 5 days or pivmecillinam 400 mg three times daily

for 3–5 days [15]. The empiric approach to uncomplicated UTI should be based on the following considerations:

- − presence of urinary symptoms in absence of vaginal discharge
- − data in local antimicrobial resistance surveillance
- − patient's previous antibiotic exposure

### 3.7. After Antibiotic Therapy in a Woman with Acute Cystitis, Is It Always Necessary to Perform a Urine Culture?

No. An uncomplicated UTI does not necessitate a urine culture unless the woman has experienced a failure of empiric therapy [15]. Moreover, performing a urine culture in asymptomatic women increases the risk to treat an asymptomatic bacteriuria.

### 3.8. What Are the Risk Factors Related to a High Risk of UTI Recurrence?

The management of patients affected by recurrent UTI is strictly related to an accurate evolution of all risk factors related to UTI recurrence. In line with the LUTIRE nomogram (Lower Urinary Tract Infections REcurrent risk) all the following risk factors have been associated with a high risk of new symptomatic UTI episode: number of partners, bowel function, type of pathogens isolated (Gram-positive/negative), hormonal status, number of previous urinary tract infection recurrences and previous treatment of asymptomatic bacteriuria [18]. Recently, Cai et al. demonstrated that risk factor identification and counseling may change the natural history of rUTI and reduce the number of symptomatic episodes and antibiotic use [6]. In this sense, the first approach for patients affected by rUTI should be the risk factors evaluation. The following risk factors have been indicated by EAU guidelines on urological infections as increasing the risk of a new episode [15] (Table 2):

**Table 2.** The table shows the age-related associations of rUTI in women—EAU guidelines [15].

| *Young and Pre-Menopausal Women* | *Post-Menopausal and Elderly Women* |
|---|---|
| Sexual intercourse | History of UTI before menopause |
| Use of spermicide | Urinary incontinence |
| A new sexual partner | Atrophic vaginitis due to estrogen deficiency |
| A mother with a history of UTI | Cystocele |
| History of UTI during childhood | Increased post-void urine volume |
| Blood group antigen secretory status | Blood group antigen secretory status |
| | Urine catheterization and functional status deterioration in elderly institutionalized women |

Finally, some authors highlight the role of blood group secretor status as a risk factor for the development of urinary tract infections and rUTI [19]. In particular, to be nonsecretors of histo-blood group antigens is a demonstrated risk factor for rUTI [19].

### 3.9. Can I Use Antibiotic Therapy as a Prophylaxis for Recurrent Cystitis, at the First Evaluation?

EAU Guidelines consider antibiotic prophylaxis for recurrent cystitis in motivated and selected patients, considering the principles of antimicrobial stewardship [15]. Cai et al. in a prospective study suggest stratifying all women affected by rUTI into three risk groups on the basis of LUTIRE nomogram findings [18] in order to evaluate the risk of a new symptomatic episode. In this sense and on the basis of recurrence risk, the patients should be treated as follows (Table 3):

**Table 3.** The table shows the therapeutic approach to rUTI patients according to the LUTIRE nomogram risk factors stratification [18].

| *Low-Risk Group* | *Moderate-Risk Group* | *High-Risk Group* |
|---|---|---|
| evaluation of risk factors | evaluation of risk factors | evaluation of risk factors |
| counseling | counseling | counseling |
| | active prophylaxis (motivate patients) | active prophylaxis (all patients) |

*3.10. What Are the Most Appropriate Strategies for Reducing the Risk of Recurrent UTI?*

In the era of antibiotic resistance, an antibiotic-sparing approach represents an important and needed alternative treatment of uncomplicated cystitis in women [9,15]. Several antibiotic-sparing approaches have been proposed with interesting results. The EAU guidelines suggest the following antibiotic-sparing approach [15]:

− Increase fluid intake
− Use immunoactive prophylaxis
− Use vaginal estrogen replacement in post-menopausal women
− Use of local or oral probiotics containing strains of proven efficacy for vaginal flora regeneration
− Use endovesical instillations of hyaluronic acid or a combination of hyaluronic acid and chondroitin sulphate
− Use continuous or post-coital antimicrobial prophylaxis when nonantimicrobial interventions have failed. Self-administered short-term antimicrobial therapy should be considered, too. Please stick to the principles of antimicrobial stewardship
− Use of cranberry products and D-mannose, even if the quality of evidence is low and there are some contradictory findings

Recently, a systematic review and meta-analysis demonstrated that a medication containing xyloglucan, hibiscus, and propolis is superior to other treatments in terms of clinical effectiveness in women affected by uncomplicated UTI and is associated with high patient compliance [20]. The authors focused their attention on the promising role of the intestinal barrier in the pathogenesis of gastrointestinal diseases and rUTI [21,22]. It is well known that uropathogenic Escherichia coli (UPEC) is the most common cause of UTI and the human bowel could be considered as a reservoir of UPEC [22]. In this sense, antibiotic-sparing approaches increasing the mechanical barrier protection have been recommended for reducing bacteria proliferation and decreasing of the load of UPEC in the bowel and decreasing the bacteria spreading into the perineal area [22].

*3.11. Which Is the Xyloglucan Mechanism of Action in Reducing Recurrent UTI?*

The clinical efficacy of xyloglucan on uncomplicated cystitis is due to the fact that xyloglucan, extracted from the seeds of the tamarind tree (Tamarindus indica), is a 'mucosal protector' with the capability to produce a defensive barrier on human epithelial tissues [23]. In vitro studies highlighted how xyloglucan could be efficacious for the management of UTIs by lowering the intestinal reservoirs of uropathogens colonizing the lower intestinal tract, as well as by interfering in the process by which microorganisms come into contact with uroepithelial cells [24]. Several clinical trials demonstrated that an antibiotic-sparing approach with xyloglucan is able to reduce antibiotic use, reduce the risk of antibiotic resistance and reduce the impact on intestinal and vaginal microbiota [20,25,26].

**4. Discussion and Final Remarks**

The management of recurrent UTIs is still a challenge and requires special attention by all physicians, due to the lack of a standard approach and due to the burden of diseases both on personal and societal aspects. The risk of collateral damages related to a wrong approach to patients affected by rUTI is high. In everyday clinical practice, the management

of recurrent UTIs should be based considering on the following considerations in order to maintain high adherence to antimicrobial stewardship principles:

- Consider the high impact of uncomplicated and recurrent UTIs on patients' quality of life (Level 3; Grade B).
- The management of recurrent UTIs should be performed in line with international guidelines (Level 1; Grade A).
- Antibiotic treatment should be used in case of urinary symptoms and absence of vaginal infection (Level 1; Grade A).
- Antibiotic treatment duration should be minimized, with the exact dosage and time schedule depending on the type of infection and in line with international guidelines recommendations (Level 1; Grade A).
- Treat asymptomatic bacteriuria only in pregnant women and before urological procedures (Level 1; Grade A).
- Before prescribing antibiotic therapy, please consider all possible collateral damages caused by antibiotics!
- After empiric therapy, please do not prescribe a urine culture unless the woman has experienced recurrence symptoms (Level 2; Grade B).
- Please consider risk factors evaluation as a pivotal step in the management of rUTI (Level 2; Grade B).
- In the era of antibiotic resistance, an antibiotic-sparing approach represents an important and needed alternative treatment of uncomplicated cystitis in women (Level 1; Grade A).

**Author Contributions:** Conceptualization, T.C.; review article research, T.C., M.L. and C.T.; writing—original draft preparation, T.C.; writing—review and editing, C.T.; All authors have read and agreed to the published version of the manuscript.

**Funding:** This research received no external funding.

**Conflicts of Interest:** The authors declare no conflict of interest.

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
