# Peer review of "Recurrent UTI: Questions and Answers on Clinical Practice"

_2673-4397, doi:10.3390/uro2040029_

Round 1
Reviewer 1 Report
tables 2 and 3 should be cited in the text of the manuscript as is Table 1,
lines 64-65 it sounds like an unfinished sentence...
line 201 bBefore prescribing... should be more appropriate
The results are based on the 12 articles>2-4,6,13,15,16,19,21,22,24,25. Yet, we have other citations too...I would recommend to seperate the results from the discussions to have clear vision on the findings of this narrative study...
there are similar references,one should consider to use the most relevant ones
Author Response
Response to the Editor
Dear Editor,
I’m submitting a revised version of the manuscript (uro-1760862), in line with the Referee’s comments. Thank you for your attention to our paper.
Reviewer #1 (Comments to the Author):
tables 2 and 3 should be cited in the text of the manuscript as is Table 1,
Response:
Many thanks for your suggestions. The table 2 and 3 have been cited in the text.
lines 64-65 it sounds like an unfinished sentence...
Response:
In line with your comments, the sentence “UTI are the most common outpatient infections in the United States (US) with an epidemiological spike in young women aged [4]” has been changed with the following “UTI are the most common outpatient infections in the United States (US) with an epidemiological spike in young women [4]”.
line 201 bBefore prescribing... should be more appropriate
Response:
The word “prescribe” has been replaced by the word “prescribing”.
The results are based on the 12 articles>2-4,6,13,15,16,19,21,22,24,25. Yet, we have other citations too...I would recommend to seperate the results from the discussions to have clear vision on the findings of this narrative study...
Response:
We totally agree with you. All study included in the results section has been included in the table 1, too. This allows to better understanding which papers have been considered for the narrative review.
there are similar references,one should consider to use the most relevant ones
Response:
The list of references has been improved in line with your suggestion.
Reviewer 2 Report
This narrative review covers a very actual topic, namely recurrent UTI management focusing on the identification of risk factors and new therapeutic strategies.
I have made a few comments, aiming to improve the manuscript quality and readability.
1. Table 1: It is suggested to provide the explanation for the abbreviations (even under or in the table): QoL, LUTIRE, because it is used first time without any explanation.
2. Line 85: References needed for “Several times, antimicrobials are prescribed inappropriately….”
3. Line 87-88: Something is wrong with the sentence, it can not be understood; there are more sentences which need improvement (English grammar)!
4. Line 93 and 197: It is suggested to include (antibiotic) agent selection next to duration and dosage
5. Line 92-94: dot is not needed at the end of the sentences
6. Line 106: do you need two of “??” at the end of the sentence? It can be used, if you want to highlight it.
7. Line 151: references are needed “several antibiotic-sparing approaches have been proposed…”
8. Line 170: E. coli is…not are!
Author Response
Response to the Editor
Dear Editor,
I’m submitting a revised version of the manuscript (uro-1760862), in line with the Referee’s comments. Thank you for your attention to our paper.
Reviewer #2 (Comments to the Author):
- Table 1: It is suggested to provide the explanation for the abbreviations (even under or in the table): QoL, LUTIRE, because it is used first time without any explanation.
Response:
The Table 1 legend has been implemented in line with your suggestions.
- Line 85: References needed for “Several times, antimicrobials are prescribed inappropriately….”
Response:
The references number 9 and 15 have been added to the following sentence: “Several times, antimicrobials are prescribed inappropriately in outpatients affected by uncomplicated UTI [9,15]”.
- Line 87-88: Something is wrong with the sentence, it can not be understood; there are more sentences which need improvement (English grammar)!
Response:
The following sentence has been deleted: “Moreover, due to the high prevalence of women affected by uncomplicated UTI, the impact of its management on antimicrobial stewardship is, then, high”.
- Line 93 and 197: It is suggested to include (antibiotic) agent selection next to duration and dosage
Response:
The sentence: “antibiotic treatment duration should be minimized, with the exact dosage and time schedule depending on the type of infection and in line with international guidelines recommendations” has been replaced by the following: “antibiotic selection, antibiotic dosage and time schedule should be selected in line with international guidelines recommendations”.
- Line 92-94: dot is not needed at the end of the sentences
Response:
The dot has been deleted.
- Line 106: do you need two of “??” at the end of the sentence? It can be used, if you want to highlight it.
Response:
One ? has been deleted.
- Line 151: references are needed “several antibiotic-sparing approaches have been proposed…”
Response:
The references number 9 and 15 have been added to the following sentence: “In the era of antibiotic resistance, an antibiotic-sparing approach represents an important and needed alternative treatment of uncomplicated cystitis in women [9,15]”.
- Line 170: E. coli is…not are!
Response:
This mistake has been corrected.
Reviewer 3 Report
At first, I would like to commend the authors for their efforts. It is a well-structured manuscript that provides important suggestions to clinicians regarding day-to-day management of UTI. I have only minor comments:
-Table 2: How does Blood group antigen secretory status increase the chances for UTI?
-Line 155- Why is the immunoactive prophylaxis being prescribed? What's the level of evidence?
-It would be better if the authors can provide the levels of evidence and grades of recommendation for each of the recommendations/suggestions throughout the article. This would help the readers to better comprehend the suggestions. Blanket statements of summary without levels of evidence bring less value whatsoever.
Author Response
Response to the Editor
Dear Editor,
I’m submitting a revised version of the manuscript (uro-1760862), in line with the Referee’s comments. Thank you for your attention to our paper.
Reviewer #3 (Comments to the Author):
At first, I would like to commend the authors for their efforts. It is a well-structured manuscript that provides important suggestions to clinicians regarding day-to-day management of UTI. I have only minor comments:
-Table 2: How does Blood group antigen secretory status increase the chances for UTI?
Response:
Many thanks for your suggestion. The role of blood group antigen status has been considered and inserted in the table 2 and results section.
-Line 155- Why is the immunoactive prophylaxis being prescribed? What's the level of evidence?
-It would be better if the authors can provide the levels of evidence and grades of recommendation for each of the recommendations/suggestions throughout the article. This would help the readers to better comprehend the suggestions. Blanket statements of summary without levels of evidence bring less value whatsoever.
Response:
Considering the EAU guideline the immunoactive prophylaxis presciption is high. However, in line with your suggestion all levels of evidence and grades of recommendation have been added to the text.
Round 2
Reviewer 1 Report
Reference 3 is added to table 1 but is not used in the results part
Author Response
Many thanks for your comment.
The reference 3 is added to the results part.
Reviewer 2 Report
Thank you very much for the corrections!
Author Response
Many thanks for your comments and suggestions.